# Submodular Meta-Learning

**Arman Adibi**
ESE Department
University of Pennsylvania
Philadelphia, PA 19104
aadibi@seas.upenn.edu

**Aryan Mokhtari**
ECE Department
University of Texas at Austin
Austin, TX 78712
mokhtari@austin.utexas.edu

**Hamed Hassani**
ESE Department
University of Pennsylvania
Philadelphia, PA 19104
hassani@seas.upenn.edu

## Abstract

In this paper, we introduce a discrete variant of the Meta-learning framework. Meta-learning aims at exploiting prior experience and data to improve performance on future tasks. By now, there exist numerous formulations for Meta-learning in the continuous domain. Notably, the Model-Agnostic Meta-Learning (MAML) formulation views each task as a continuous optimization problem and based on prior data learns a suitable initialization that can be adapted to new, unseen tasks after a few simple gradient updates. Motivated by this terminology, we propose a novel Meta-learning framework in the discrete domain where each task is equivalent to maximizing a set function under a cardinality constraint. Our approach aims at using prior data, i.e., previously visited tasks, to train a proper initial solution set that can be quickly adapted to a new task at a relatively low computational cost. This approach leads to (i) a personalized solution for each task, and (ii) significantly reduced computational cost at test time compared to the case where the solution is fully optimized once the new task is revealed. The training procedure is performed by solving a challenging discrete optimization problem for which we present deterministic and randomized algorithms. In the case where the tasks are monotone and submodular, we show strong theoretical guarantees for our proposed methods even though the training objective may not be submodular. We also demonstrate the effectiveness of our framework on two real-world problem instances where we observe that our methods lead to a significant reduction in computational complexity in solving the new tasks while incurring a small performance loss compared to when the tasks are fully optimized.

## 1 Introduction

Many applications in artificial intelligence necessitate exploiting prior data and experience to enhance quality and efficiency on new tasks. This is often manifested through a set of tasks given in the training phase from which we can learn a model or representation that can be used for new unseen tasks in the test phase. In this regard, Meta-learning aims at exploiting the data from the available tasks to learn model parameters or representation that can be later used to perform well on new unseen tasks, in particular, when we have access to limited data and computational power at the test time [1–4]. By now, there are several formulations for Meta-learning, but perhaps one of the most successful ones is the Model-Agnostic Meta-Learning (MAML) [5]. In MAML, we aim to train the

model parameters such that applying a few steps of gradient-based updates with a small number of samples from a new task would perform well on that task. MAML can also be viewed as a way to provide a proper initialization, from which performance on a new task can be optimized after a few gradient-based updates. Alas, this scheme only applies to settings in which the decision variable belongs to a continuous domain and can be adjusted using gradient-based methods at the test time.

Our goal is to extend the methodology of MAML to the discrete setting. We consider a setting that our decision variable is a discrete set, and our goal is to come up with a good initial set that can be quickly adjusted to perform well over a wide range of new tasks. In particular, we focus on submodular maximization to represent the tasks which is an essential class of discrete optimization.

There are numerous applications where the submodular meta-learning framework can be applied to find a personalized solution for each task while significantly reducing the computation load. In general, most recommendation tasks can be cast as an instance of this setting [6–8]. Consider the task of recommending a set of items, e.g., products, locations, ads, to a set of users. One approach for solving such a problem is to find the subset of items that have the highest score over all the previously-visited users and recommend that subset to a new user. Indeed, this approach leads to a reasonable performance at test time; however, it does not provide a user-specific solution for a new user. Another approach is to find the whole subset at the test time when the new user arrives. In contrast to the previous approach, this scheme leads to a user-specific solution, but at the cost of running a computationally expensive algorithm to select all the elements at the test time.

In our Meta-learning framework, the process of selecting set items to be recommended to a new user is done in two parts: In the first part, a set of items are selected offline according to prior experience. These are the most popular items to the previously-visited users (depending on the context). In the second part, which happens at the test time, a set of items that is *personalized* to the coming user is selected. These are items that are computed specifically according to the features of the coming user. In this manner, the computation for each coming user would be reduced to the selection of the second part, which typically constitutes a small portion of the final set of recommended items. The first part can be done offline with a lower frequency. For instance, in a real recommender system, the first part can be computed once every hour, and the second part can be computed specifically for each coming user (or for a class of similar users). While we have mentioned recommendation (or more generally facility location) as a specific example, it is easy to see that this framework can be easily used to reduce computation in other notable applications of submodular optimization.

**Contributions.** Our contributions are threefold:

- We propose a novel discrete Meta-learning framework where each task is equivalent to maximizing a set function under some cardinality constraint. Our framework aims at using prior data, i.e., previously visited tasks, to train a proper initial solution set that can be quickly adapted to a new task at a low computational cost to obtain a task-specific solution.
- We present computationally efficient deterministic and randomized meta-greedy algorithms to solve the resulting meta-learning problem. When the tasks are monotone and submodular, we prove that the solution obtained by the deterministic algorithm is at least $0.53$-optimal, and the solution of the randomized algorithm is $(1 - 1/e - o(1))$-optimal in expectation, where the $o(1)$ term vanishes by the size of the solution. These guarantees are obtained by introducing new techniques, despite that the meta-learning objective is *not* submodular.
- We study the performance of our proposed meta-learning framework and algorithms for movie recommendation and ride-sharing problems. Our experiments illustrate that the solution of our proposed meta-learning scheme, which chooses a large portion of the solution in the training phase and a small portion adaptively at test time, is very close to the solution obtained by choosing the entire solution at the test time when a new task is revealed.

## 1.1 Related work

**Continuous Meta-Learning.** Meta-learning has gained considerable attention recently mainly due to its success in few shot learning [9–12] as well as reinforcement learning [13–15]. One of the most successful forms of meta-learning is the gradient-based *Model Agnostic Meta-learning* (MAML) approach [5]. MAML aims at learning an initialization that can be adapted to a new task after performing one (or a few) gradient-based update(s); see, e.g., [16]. This problem can be written as

$$\min_{w \in W} \mathbb{E}_{a \sim P}[f_a(w - \nabla f_a(w))],  \tag{1}$$

where $W \subseteq \mathbb{R}^d$ is the feasible set and $P$ is the probability distribution over tasks. The previous works on MAML including [16–22] consider the case where $W$ is a continuous space. In fact none of these works can be applied to the case where the feasible parameter space is discrete. In this paper, we aim to close this gap and extend the terminology of MAML to discrete settings.

**Submodular Maximization.** Submodular functions have become key concepts in numerous applications such as data summarization [23–26], viral marketing [27], sensor placement [28], dictionary learning [29], and influence maximization [27]. It is well-known that for maximizing a monotone and submodular function under the cardinality constraint, the greedy algorithm provides a $(1 - 1/e)$-optimal solution [30–32]. There has been significant effort to improve the scalability and efficiency of the greedy algorithm using lazy, stochastic, and distributed methods [33–38]. However, our framework is fundamentally different and complementary to these approaches as it proposes a new approach to use data at training time to improve performance at new tasks. Indeed, all the aforementioned techniques can be readily used to further speed-up our algorithms. Optimization of related submodular tasks has been a well-studied problem with works on structured prediction [39], submodular bandits [8, 40], online submodular optimization [41–43], and public-private data summarization [44]. However, unlike our work, these approaches are not concerned with train-test phases for optimization. Another recently-developed methodology to reduce computation is the two-stage submodular optimization framework [45–47], which aims at summarizing the ground set to a reasonably small set that can be used at test time. The main difference of our framework with the two-stage approaches is that we allow for *personalization*: A small subset of items that can be found at test time specific to the task at hand. This leads to a completely new problem formulation, and consequently, new algorithms (more details in the supplementary material).

## 2 Problem Statement: Discrete Meta-Learning

**Setup.** We consider a family of tasks $\mathcal{T} = \{\mathcal{T}_i\}_{i \in \mathcal{I}}$, where the set $\mathcal{I}$ could be of infinite size. Each task $\mathcal{T}_i$ is represented via a set function $f_i : 2^V \to \mathbb{R}_+$ that measures the reward of a set $S \subseteq V$ for the $i$-th task, and performing the task $\mathcal{T}_i$ would mean to maximize the function $f_i$ subject to a given constraint. For instance, in a recommender system where we aim to recommend a subset of the items to the users, the set $\mathcal{I}$ denotes the set of all the possible users and selecting which items to recommend to a user $i \in \mathcal{I}$ is viewed as the task $\mathcal{T}_i$. Moreover, the function $f_i$ encodes the users satisfaction, i.e., $f_i(S)$ quantifies how suitable the set of items $S$ is for user $i$. Taking a statistical perspective, we assume that the tasks $\mathcal{T}_i$ occur according to a possibly unknown probability distribution $i \sim p$.

In this paper, we focus on the case where the functions $f_i$ are monotone and submodular set functions and each task $\mathcal{T}_i$ amounts to maximizing $f_i$ under the $k$-cardinality constraint. That is, the task $\mathcal{T}_i$ is to select a subset $S \subseteq V$ of size $k$ such that the value of $f_i(S)$ is maximized. Submodularity of $f_i$ means that for any $A, B \subseteq V$ we have $f_i(A) + f_i(B) \geq f_i(A \cup B) + f_i(A \cap B)$. Furthermore, $f_i$ is called monotone if for any $A \subseteq B$ we have $f_i(A) \leq f_i(B)$.

**Training and test tasks.** We assume access to a collection of *training* tasks $\{\mathcal{T}_i\}_{i=1}^m$. These are the tasks that we have already experienced, i.e., they correspond to the users that we have already seen. Formally, this means that for each training task $\mathcal{T}_i$, we assume knowledge of the corresponding function $f_i$. In our formulation, each of the training tasks is assumed to be generated i.i.d. according to the distribution $p$. Indeed, eventually we aim to optimize performance at *test* time, i.e., obtain the best performance for new and unseen tasks generated independently from the distribution $p$. For instance, in our recommendation setting, test tasks correspond to new users that will arrive in the future. Our goal is to use the training tasks to reduce the computation load at test time.

**Two extremes of computation.** Let us use $\mathcal{T}_{\text{test}}$ (and $f_{\text{test}}$) to denote the task (and its corresponding set function) that we aim to learn at test time. Ideally, if we have sufficient computational power, then we should directly optimize $f_{\text{test}}$ by solving the following problem

$$\max_{S \in V, |S| \leq k} f_{\text{test}}(S). \tag{2}$$

We denote the optimal solution of (2) by $S_{\text{test}}^*$. For instance, we can use the greedy procedure to solve (2) which leads to a $(1 - 1/e)$-optimal solution using $\mathcal{O}(kn)$ evaluations of $f_{\text{test}}$, and through $k$ passes over the ground set. However, the available computational power and time in the test phase is often limited, either because we need to make quick decisions to respond to new users or since we need to save energy. For instance, in real-world advertising or recommendation systems, both these requirements are crucial: many users arrive within each hour which means fast optimization is crucial

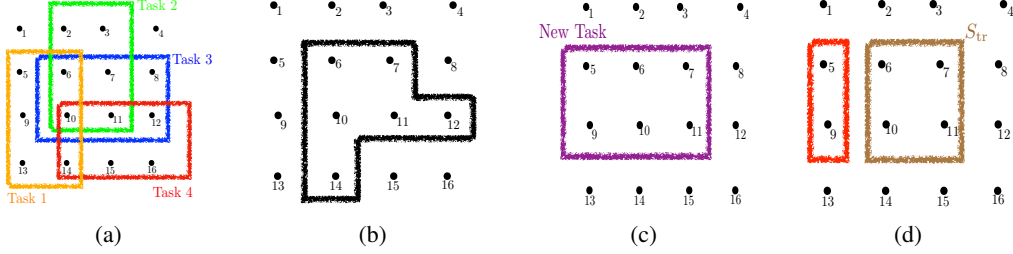

Figure 1: (a) Optimal sets for each of the training tasks ($k = 6$); (b) the set obtained by solving the average problem in (4); (c) the optimal set for a new task revealed at test time, i.e. solving (2); (d) the optimal set for the new task is also obtained by solving the Meta-learning problem in (7) with $l = 4$ (brown set) and adding the task-specific elements at test time (red set).

(especially if $n, k$ are large), and also, reducing computation load would lead to huge energy savings in the long run. In such cases, Problem (2) should be solved approximately with less computation.

An alternative to reduce computation at test time is to solve the problem associated with the expected reward over all possible tasks in the training phase (when we have enough computation time), i.e.,

$$\max_{S \in V, |S| \leq k} \mathbb{E}_{i \sim p} \left[ f_i(S) \right]. \tag{3}$$

We denote the optimal solution of (3) by $S^*_{\exp}$. The rationale behind this approach is that the optimal solution to this problem would generalize well over an unseen task if the new task is also drawn according to the probability distribution $p$. In other words, the solution of (3) should perform well for the problem in (2) that we aim to solve at the test time, assuming that $f_{\text{test}}$ is sampled according to $p$. In this way, we do not need any extra computation at the test time. However, in this case, the solution that we obtain would not be the best possible solution for the task that we observe at the test time, i.e., $S^*_{\text{test}}$ is not equal to $S^*_{\exp}$. As we often do not have access to the underlying probability distribution $p$, and we only have access to a large set of realizations of tasks in the training phase. As a result, instead of solving (3), we often settle for maximizing the sample average function

$$\max_{S \in V, |S| \leq k} \frac{1}{m} \sum_{i=1}^{m} f_i(S), \tag{4}$$

where $m$ is the number of available tasks in the training phase.

Problems (2) and (4) can be considered as two different extreme cases. In the first option, by solving (2), we avoid any pre-processing in the training phase, and we obtain the best possible guarantee for the new task, but at the cost of performing computationally expensive operations (e.g., full greedy) at the test time. In the second approach, by solving (4) in the training phase, we obtain a solution that possibly performs reasonably without any computation at the test phase, but the quality of the solution may not be as good as the first option. In summary, there exists a trade-off between the required computational cost at the test time and the performance guarantee on the unseen task. Hence, a fundamental question that arises is what would be the best scheme at the training phase assuming that at test time we have some limited computational power. For instance, in the monotone submodular case, assume that instead of running the greedy algorithm for $k$ rounds, which has a complexity $\mathcal{O}(kn)$, we can only afford to run $\alpha k$ rounds of greedy at test time, which has complexity $\mathcal{O}(\alpha nk)$, where $\alpha \in (0, 1)$ is small. In this case, a natural solution would be to find an appropriate set of $(1 - \alpha)k$ elements in the training phase, and add the remaining $\alpha k$ elements at test time when a new task arrives. This discussion also applies to any other greedy method (e.g., lazy or stochastic greedy).

**Discrete Meta-Learning.** As we discussed so far, when computational power is limited at test time, it makes sense to divide the process of choosing the best decision between training and test phases. To be more specific, in the training phase, we choose a subset of elements from the ground set that would perform over the training tasks, and then select (or optimize) the remaining elements at the test time *specifically* with respect to the task at hand. To state this problem, consider $S_{\text{tr}} \subseteq V$ with cardinality $|S_{\text{tr}}| = l$, where $l < k$, as the initial set that we aim to find at the training phase, and the set $S_i$ that we add to the initial set $S_{\text{tr}}$ at test time (See Figure 1 for an illustration). Hence, the problem of interest can be written as

$$\max_{S_{\text{tr}} \in V, |S_{\text{tr}}| \leq l} \mathbb{E}_{i \sim p} \left[ \max_{S_i \in V, |S_i| \leq k-l} f_i(S_{\text{tr}} \cup S_i) \right], \tag{5}$$

Note that the critical decision variable that we need to find is $S_{\text{tr}}$ which is the best initial subset of size $l$ overall all possible choices of task when a best subset of size $k - l$ is added to that. In fact, if we define $f_i'(S_{\text{tr}}) := \max_{S_i \in V, |S_i| \le k-l} f_i(S_{\text{tr}} \cup S_i)$, then we can rewrite the problem in (5) as

$$\max_{S_{\text{tr}} \in V, |S_{\text{tr}}| \le l} \mathbb{E}_{i \sim p} \left[ f_i'(S_{\text{tr}}) \right]. \tag{6}$$

As described previously, we often do not have access to the underlying probability distribution $p$ of the tasks, and we instead have access to a large number of sampled tasked that are drawn independently according to $p$. Hence, instead of solving (5), we solve its sample average approximation given by

$$\max_{S_{\text{tr}} \in V, |S_{\text{tr}}| \le l} \frac{1}{m} \sum_{i=1}^{m} \left[ \max_{S_i \in V, |S_i| \le k-l} f_i(S_{\text{tr}} \cup S_i) \right] = \max_{S_{\text{tr}} \in V, |S_{\text{tr}}| \le l} \frac{1}{m} \sum_{i=1}^{m} \left[ f_i'(S_{\text{tr}}) \right], \tag{7}$$

where $m$ is the number of tasks in the training set which are sampled according to $p$. Even though the functions $f_i$ are submodular, $f_i'$ *is not submodular* or $k$-submodular [48] (see the supplementary materials for specific counter examples). Hence, Problem (7) is not a submodular maximization problem. In the next section, we present algorithms for solving Problem (7) with provable guarantees.

We finally note that Problem (7) will be solved at *training* time to find the solution $S_{\text{tr}}$ of size $l$. This solution is then *completed at test time*, by e.g. running $k - l$ further rounds of greedy on the new task, to obtain a task-specific solution of size $k$.

## 3 Algorithms for Discrete Submodular Meta-Learning

Solving Problem (7) requires finding a set $S_{\text{tr}}$ for the outer maximization and sets $\{S_i\}_{i=1}^{m}$ for the inner maximization. In this section, we describe our proposed greedy-type algorithms to select the elements $S_{\text{tr}}$ and $\{S_i\}_{i=1}^{m}$. As we deal with $m + 1$ sets, the order in which the sets $S_{\text{tr}}$ and $\{S_i\}_{i=1}^{m}$ are updated becomes crucial, i.e., it is not clear which of the sets $S_{\text{tr}}$ or $S_i$'s should be preferably updated in each round and how can the functions $f_i$ be incorporated in finding the right order, which is the main challenge in designing greedy methods to solve (7). We design greedy procedures with both deterministic and randomized orders and provide strong guarantees for their solutions.

### 3.1 Deterministic Algorithms

In this section, we first describe Algorithms 1 and 2 which use specific orderings to solve Problem (7). Based on these two, we then design Algorithm 3 as our *main deterministic* algorithm. Throughout this section, we use $\Delta_i(e|S) = f_i(S \cup \{e\}) - f_i(S)$ to denote the marginal gain of adding an element $e$ to set $S$ for function $f_i$. In brief, Algorithm 1 first fills $S_{\text{tr}}$ greedily up to completion and then it constructs each of the $S_i$'s greedily on the top of $S_{\text{tr}}$. Specifically, starting from the empty set initialization for $S_{\text{tr}}$ and $S_i$'s, Algorithm 1 constructs in its first phase the set $S_{\text{tr}}$ in $l$ rounds, by adding one element per round, where the next element in each round is chosen according to $e^* = \arg\max_{e \in V} \sum_{i=1}^{m} f_i(S_{\text{tr}} \cup \{e\}) - f_i(S_{\text{tr}})$. Once $S_{\text{tr}}$ is completed, in the second phase, each of the sets $S_i$ is constructed in parallel by running the greedy algorithm on $f_i$. That is, each $S_i$ is updated in $k - l$ rounds where in each round an element with maximum marginal on $f_i$ is added to $S_i$ based on $e_i^* = \arg\max_{e \in V} f_i(S_{\text{tr}} \cup S_i \cup \{e\}) - f_i(S_{\text{tr}} \cup S_i)$.

Algorithm 2 uses the opposite ordering of Algorithm 1. Initializing with all sets to be empty, in the first phase it constructs the sets $S_i$ using the greedy procedure on $f_i$, i.e., each $S_i$ is updated in parallel in $k - l$ rounds, where in each round the element $e_i^* = \arg\max_{e \in V} f_i(S_{\text{tr}} \cup S_i \cup \{e\}) - f_i(S_{\text{tr}} \cup S_i)$ is added to $S_i$. In the second phase, the set $S_{\text{tr}}$ is formed greedily in $l$ rounds, and in each round the following element is added $e^* = \arg\max_{e \in V} \sum_{i=1}^{m} f_i(S_{\text{tr}} \cup \{e\} \cup S_i) - f_i(S_{\text{tr}} \cup S_i)$.

While the solutions obtained by Algorithms 1 and 2 are guaranteed to be near-optimal, it turns out that they can be complementary with respect to each other. Our *main* deterministic algorithm, called `Meta-Greedy`, runs both Algorithms 1 and 2 and chooses as output the solution, among the two, that leads to a higher objective value in (7). Next, we explain why our `Meta-Greedy` method can outperform both Algorithms 1 and 2. This will be done by providing the theoretical guarantees for these methods and consequently explaining why Algorithms 1 and 2 are complementary.

**Theoretical guarantees.** We begin with the analysis of Algorithm 1. The following proposition relates the overall performance of Algorithm 1 to its performance after phase 1 and shows that the output of the algorithm is at least $1/2$-optimal. We use OPT for the optimal value of Problem (7).

**Algorithm 1**

1: **Initialize** $S_{\mathrm{tr}} = \{S_i\}_{i=1}^m = \emptyset$
   /* Phase 1: */
2: **for** $t = 1, 2, \ldots, l$ **do**
3:     Find $e^* = \arg\max_{e \in V} \sum_{i=1}^m \Delta_i(e|S_{\mathrm{tr}})$
4:     $S_{\mathrm{tr}} \leftarrow S_{\mathrm{tr}} \cup \{e^*\}$
5: **end for**
   /* Phase 2: */
6: **for** $t = 1, 2, \ldots, k - l$ **do**
7:     **for** $i = 1, 2, \ldots, m$ **do**
8:         Find $e_i^* = \arg\max_{e \in V} \Delta_i(e|S_{\mathrm{tr}} \cup S_i)$
9:             $S_i \leftarrow S_i \cup \{e_i^*\}$
10:     **end for**
11: **end for**
12: Return $S_{\mathrm{tr}}$ and $\{S_i\}_{i=1}^m$

**Algorithm 2**

1: **Initialize** $S_{\mathrm{tr}} = \{S_i\}_{i=1}^m = \emptyset$
   /* Phase 1: */
2: **for** $i = 1, 2, \ldots, m$ **do**
3:     **for** $t = 1, 2, \ldots, k - l$ **do**
4:         Find $e_i^* = \arg\max_{e \in V} \Delta_i(e|S_i)$
5:             $S_i \leftarrow S_i \cup \{e_i^*\}$
6:     **end for**
7: **end for**
   /* Phase 2: */
8: **for** $t = 1, 2, \ldots, l$ **do**
9:     Find $e^* = \arg\max_{e \in V} \sum_{i=1}^m \Delta_i(e|S_{\mathrm{tr}} \cup S_i)$
10:     $S_{\mathrm{tr}} \leftarrow S_{\mathrm{tr}} \cup \{e^*\}$
11: **end for**
12: Return $S_{\mathrm{tr}}$ and $\{S_i\}_{i=1}^m$

---

**Algorithm 3** `Meta-Greedy`

1: Run Algorithms 1 and 2 and obtain respective solution sets $S_{\mathrm{tr}}^{(1)}, \{S_i^{(1)}\}_{i=1}^m$ and $S_{\mathrm{tr}}^{(2)}, \{S_i^{(2)}\}_{i=1}^m$.
2: Compute the objective value $\sum_{i=1}^m f_i(S_{\mathrm{tr}} \cup S_i)$ for both solution sets.
3: Return $S_{\mathrm{tr}}$ and $S_i$ of the solution set that has a higher objective value.

---

**Proposition 1.** *Let $S_{\mathrm{tr}}, \{S_i\}_{i=1}^m$ be the output of Algorithm 1, and define $\beta$ as $\beta := \sum_{i=1}^m f_i(S_{\mathrm{tr}})$. If the functions $f_i$ are monotone and submodular, then*

$$\sum_{i=1}^m f_i(S_{\mathrm{tr}} \cup S_i) \geq \max\left\{\beta,\, (1 - 1/e)(\mathrm{OPT} - 2\beta) + \beta\right\}.$$

*Consequently, the solution obtained by Algorithm 1 is at least $1/2$-optimal for any value of $\beta$.*

The proof of this proposition is relegated to the supplementary material. The key step in the proof is to relate the progress made in phase 1 to the gap to OPT. This is indeed challenging as phase 1 only involves updates on the outer maximization of (7). In this regard, we prove a novel technical lemma that can be generally applicable to any mini-max submodular problem. The guarantee given in Proposition 1 is minimized when $\beta = \mathrm{OPT}/2$. If $\beta$ is small (e.g., $\beta = 0$) or if $\beta$ is large (e.g. if $\beta = (1 - 1/e)\mathrm{OPT}$) then the guarantee becomes tight (e.g. $(1 - 1/e)\mathrm{OPT}$). This is indeed expected from the greedy nature of the two phases of Algorithm 1. What is non-trivial about the result of Proposition 1 is that it provides a strong guarantee for any value of $\beta$, and not just cases that $\beta$ is small or large. Similarly, we can provide near-optimality guarantees for Algorithm 2.

**Proposition 2.** *Let $S_{\mathrm{tr}}, \{S_i\}_{i=1}^m$ be the output of Algorithm 2, and define $\gamma$ as $\gamma := \sum_{i=1}^m f_i(S_i)$. If the functions $f_i$ are monotone and submodular, then*

$$\sum_{i=1}^m f_i(S_{\mathrm{tr}} \cup S_i) \geq \max\left\{\gamma,\, (1 - 1/e)(\mathrm{OPT} - 2\gamma) + \gamma\right\}.$$

*Consequently, the solution obtained by Algorithm 2 is at least $1/2$-optimal for any value of $\gamma$.*

Similarly, we can show that $\gamma = \mathrm{OPT}/2$ leads to (the worst) guarantee $1/2$-OPT, while for large and small values of $\gamma$ the bound in Proposition 2 approaches the optimal approximation $(1 - 1/e)\mathrm{OPT}$.

We note that the values $\beta$ in Proposition 1 (Algorithm 1) and $\gamma$ in Proposition 2 (Algorithm 2) represent two different extremes. The value $\beta$ represents how significant is the role of the set $S_{\mathrm{tr}}$ in solving Problem (7), and $\gamma$ represents how significant the role of the sets $\{S_i\}_{i=1}^m$ can be. Even though the worst-case guarantees of Propositions 1 and 2 are obtained when $\beta, \gamma = \mathrm{OPT}/2$, a coupled analysis of the algorithms show that in this case at least one of the algorithms should output a solution which is strictly better than $1/2$-optimal. In other words, the outcomes of Algorithms 1

---

**Algorithm 4** `Randomized meta-Greedy`

---

1: **Initialize** the sets $S_{\text{tr}}$ and $\{S_i\}_{i=1}^m$ to the empty set.
2: **while** $|S_i| < k - l$ and $|S_{\text{tr}}| < l$ **do**
3:      $e_i^* \leftarrow \arg\max_{e \in V} f_i(S_{\text{tr}} \cup S_i \cup \{e\}) - f_i(S_{\text{tr}} \cup S_i)$
4:      $e_{tr}^* \leftarrow \arg\max_{e \in V} \sum_{i=1}^m f_i(S_{\text{tr}} \cup S_i \cup \{e\}) - f_i(S_{\text{tr}} \cup S_i)$
5:      **w.p.** $\frac{l}{k}$: $S_{\text{tr}} = S_{\text{tr}} \cup \{e_{tr}^*\}$
6:      **w.p.** $\frac{k-l}{k}$: $S_i = S_i \cup \{e_i^*\}, \forall i = 1, \cdots, m$
7: **end**
8: If $S_{\text{tr}}$ or $S_i$'s have not reached their cardinality limit then fill them greedily until it is reached
9: Return $S_{\text{tr}}$ and $\{S_i\}_{i=1}^m$

---

and 2 are dependent to one another, and the best performance is achieved when the maximum of the two is considered. This justifies why our main algorithm `Meta-Greedy` can perform strictly better than each of the Algorithms 1 and 2. Using a coupled analysis of the outcome of Algorithms 1 and 2, we can bound the performance of `Meta-Greedy` for different values of $\beta$ and $\gamma$ (see the proof of Theorem 1 in the supplementary materials). In particular, we can show that the output of `Meta-Greedy` is at least $0.53$-optimal. The proof of the following theorem carefully analyzes the interplay between the role of the inner and outer maximization problems in (7). We emphasize that the proof introduces new techniques applicable to other types of minimax submodular problems.

**Theorem 1.** *Consider the* `Meta-Greedy` *algorithm outlined in Algorithm 3. If the functions $f_i$ are monotone and submodular, then we have*

$$\max\Big\{\sum_{i=1}^m f_i(S_{\text{tr}}^{(1)} \cup S_i^{(1)}), \ \sum_{i=1}^m f_i(S_{\text{tr}}^{(2)} \cup S_i^{(2)})\Big\} \geq 0.53 \times \text{OPT}. \tag{8}$$

### 3.2 Randomized Algorithm

In this section, we consider greedy procedures in which the decision to alternate between the set $S_{\text{tr}}$ (the outer maximization) and the sets $\{S_i\}_{i=1}^m$ (the inner maximization) is done based on a randomized scheme. The `Randomized meta-Greedy` procedure, outlined in Algorithm 4, provides a specific randomized order. In each round, with probability $l/k$ we choose to perform a greedy update on $S_{\text{tr}}$, and with probability $1 - l/k$ we choose to perform a greedy update on *all* the $S_i$'s, $i = 1, \cdots, m$. This procedure continues until either $S_{\text{tr}}$ or $\{S_i\}_{i=1}^m$ hit their corresponding carnality constraint, in which case we continue to update the other set(s) greedily until they also become full.

The randomized update of Algorithm 4 is designed to optimally connect the expected increase the objective value at each round with the gap to OPT (as shown in the proof of Theorem 2). Hence, the `Randomized meta-Greedy` procedure is able to achieve in expectation a guarantee close to the tight value $(1 - 1/e)$OPT. However, due to the randomized nature of the algorithm, the sets $S_{\text{tr}}$ or $S_i$ might hit their carnality constraint earlier than expected. Analyzing the function value at this "stopping time" is another technical challenge that we resolve in the following theorem to obtain a guarantee that becomes slightly worse than $(1 - 1/e)$OPT depending on the values $k - l$ and $l$.

**Theorem 2.** *Let the (random) sets $S_{\text{tr}}$, $\{S_i\}_{i=1}^m$ be the output of Algorithm 4. If the functions $f_i$ are monotone and submodular, then*

$$\mathbb{E}\Big[\sum_{i=1}^m f_i(S_{\text{tr}} \cup S_i)\Big] \geq (1 - b - \exp(-1 + c))\,\text{OPT},$$

*where $b, c \to 0$ as $k - l$ and $l$ grow. More precisely, we have $b = \max\{\frac{1}{k-l}, \frac{1}{l}\}$, and $c = 3\sqrt{b \log \frac{1}{b}}$.*

**Remark 1.** *All presented algorithms are designed for the training phase and their output is the set $S_{\text{tr}}$ with size $l$. The sets $\{S_i\}_{i=1}^m$ are only computed for algorithmic purposes. Given a new task at the test phase, the remaining $k - l$ task-specific elements will be added to $S_{\text{tr}}$ using e.g. greedy updates that require a total complexity of $O((k - l)n)$ in function evaluations. Also, the training complexity of the proposed algorithms is $O(kmn)$, however, certain phases can be implemented in parallel.*

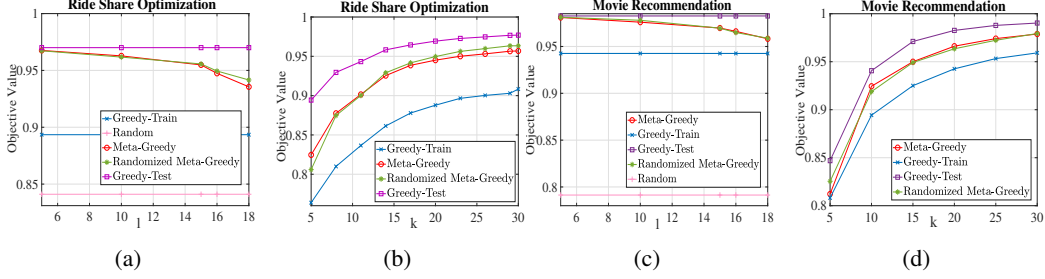

Figure 2: Performance for Ride Share Optimization (a)-(b) and Movie Recommendation (c)-(d).

## 4    Simulation Results

We provide two experimental setups to evaluate the performance of our proposed algorithms and compare with other baselines. Each setup involves a different set of tasks which are represented as submodular maximization problems subject to the $k$-cardinality constraint. We have considered the following algorithms: **Meta-Greedy** (Algorithm 3), **Randomized Meta-Greedy** (Algorithm 4), **Greedy-Train** (which chooses all the $k$ elements during the training phase–see (4) and the discussion therein), **Greedy-Test** (which chooses all the $k$ elements during the test phase–see (2) and the discussion therein), and **Random** (which chooses a random set of $k$ elements). In the following, we briefly explain the data and tasks and refer the reader to the supplementary materials for more details.

**Ride Share Optimization.** We will formalize and solve a facility location problem on the Uber dataset [49]. Our experiments were run on the portion of data corresponding to Uber pick-ups in Manhattan in the period of September 2014. This portion consists of $\sim 10^6$ data points each represented as a triplet $(latitude, longitude, DateTime)$. A customer and a driver are specified through their locations on the map. We use $u = (x_u, y_u)$ for a customer a and $r = (x_r, y_r)$ for a driver. We define the "convenience score" of a (customer, driver) pair as $c(u, r) = 2 - \frac{2}{1+e^{-200d(u,r)}}$, where $d(u, r)$ denotes the Manhattan distance [46]. Given a specific time $a$, we define a time slot $T_a$ and picking inside the data set 10 points in half an hour prior to time $a$, and for each point we further pick 10 points in its 1 km neighborhood, which makes a total of 100 points (locations) on the map. A task $\mathcal{T}_i$, takes place at a corresponding time $a_i$, and by defining the set of locations $T_{a_i}$ as above, we let $f_i$ be a monotone submodular function defined over a set $S$ of driver locations as $f_i(S) = \sum_{u \in T_{a_i}} \max_{r \in S} c(u, r)$. We pick 100,000 locations at random from the September 2014 Uber pick-up locations as a ground set. For training we form $m = 50$ tasks by picking for each task a random time in the *first* week of Sept. 2014. We test on $m = 50$ new tasks formed similarly from the *second* week of Sept. 2014 and report in the figures the average performance obtained at test tasks.

Figures 2a and 2b show the performance of our proposed algorithms against the baselines mentioned above. Figure (2a) shows the performance of all algorithms when we fix $k = 20$, and vary $l$ from 5 to 18. Larger $l$ means less computation at test time (since we need to further choose $k - l$ elements at test). However, we see that even for large values of $l$ (e.g. $l = 16$), the performance of Meta-Greedy is still quite close to the ideal performance of Greedy-Test. Putting this together with the fact that the performance of Greedy-Train is not so good, we can conclude that adding a few personalized elements at test time significantly boosts performance to be even close to the ideal. In Figure (2b), we compare the performance of all the algorithms when $k$ changes from 5 to 30, and $l$ is $80\%$ of $k$ ($l = \lfloor 0.8k \rfloor$). As we can see, even when we just learn $20\%$ of the set in test time, the performance of Meta-greedy is close to Test-Greedy. Also, when $k - l$ increases, Random-Meta-Greedy performs better than Meta-Greedy. This is in compliance with the results of Theorems 1, 2.

**Movie Recommendation.** In this application, we use the Movielens dataset [50] which consists of $10^6$ ratings (from 1 to 5) by 6041 users for 4000 movies. We pick the 2000 most rated movies, and 200 users who rated the highest number of movies (similar to [47]). We partitioned the 200 users into 100 users for the training phase and 100 other users for the test phase. Each movie can belong to one of 18 genre. For each genre $t$ we let $G_t$ be the set of all movies with in genre $t$. For each user $i$, we let $R_i$ be the set of all movie rated by the user, and for each movie $v \in R_i$ the corresponding rating is denoted by $r_i(v)$. Furthermore, for user $i$ we define $f_i(S) = \sum_{t=1}^{18} w_{i,t} \cdot \max_{v \in R_i \cap G_t \cap S} r_i(v)$ which is the weighted average over maximum rate that user $i$ gives to movies from each genre and

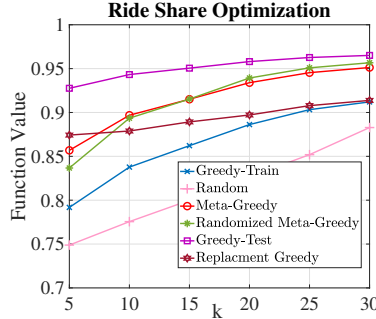

Figure 3: Comparison of two-stage framework and submodular meta-learning framework

$w_{i,t}$ is proportion of movies in genre $t$ which is rated by user $i$ out of all the rating he provides. A task $\mathcal{T}_i$ involves 5 users $i_1, \cdots, i_5$ and the function assigned to the task is the average of $f_{i_1}, \cdots, f_{i_5}$. We formed $m = 50$ training tasks from the users in the training phase, and $m = 50$ test tasks from the users in the test phase. Figure (2c) (resp. 2d) has been obtained in a similar format as Figure 2a (resp. Figure 2b). Indeed, we observe a very similar pattern as in the rideshare experiments.

**Comparison with Two-stage Submodular Optimization.** Two-stage submodular optimization is another way to deal with limited computational power in test time. In this framework, at the training time, a reduced ground set will be learned which will be used as a ground set at test time. This procedure will reduce the computational time in the test time. More formally, the two-stage submodular optimization framework aims to solve the following problem: Let $f_i : 2^{\mathcal{X}} \rightarrow \mathbb{R}_+$ for $i \in [m]$, be a monotone submodular function over ground set $V$. The goal is to find $S$ with size at most $q$ whose subsets of size $k$ maximize the sum of $f_i$ for $i \in [m]$:

$$\max_{S \subseteq \mathcal{X}, |S| \leq q} \sum_{i=1}^{m} \max_{S_i \subseteq S, |S_i| \leq k} f_i(S_i) \tag{9}$$

Once the set $S$ is found, it will be used at the test time (e.g. by running full greedy on $S$ as the reduced ground set) to find $k$ elements for a new task. Although this framework uses $\mathcal{O}(qk)$ function evaluations for each new test task, however, it poorly personalizes to a test task because the set $S$ has been optimized only for the tasks at the training time. This intuition is indeed consistent with our experimental findings reported below. We further remark that the two-stage framework needs high computational power in training; Hence, we were not able to run the state-of-the-art two-stage algorithms to solve (9) in the setting considered in our main simulation results, e.g., for a ground set of size $n = 10^5$ the implementation of two-stage approach would take a very long time.

We consider the ride-sharing application and let $n = 500$ (ground set size), $m = 50$ (number of tasks), and $k$ changing from 5 to 30 (cardinality constraint) while $l = 80\% k$ (portion that will fill in the submodular meta-learning during training), and $q = 100$ (size of reduced ground set for two-stage framework). For solving the two-stage problem (9) we have used the Replacement-Greedy algorithm introduced in [47]. We choose these parameters based on the following two facts: First, because of the high computational cost of the Replacement Greedy algorithm in training for the ride-sharing application, we chose $n$ to be 500. Second, we provide a fair comparison in terms of computational power at test time, which means both Meta-Greedy (our algorithm) and Replacement-Greedy have exactly *the same computational cost* at test time. Formally, $n(k - l) = qk$.

we report the result for the above setting in the Figure 3. A few comments are in order: (i) The two stage implementation reduces the ground set of size $n = 500$ to $q = 100$. When $k$ is small, some of the popular elements found at training time would be good enough to warrant a good performance at test time. However, when $k$ increases, the role of personalizing becomes more apparent. As we see, the performance of Replacement-Greedy does not improve much when we increase $k$ and it is close to the performance of Greedy-Train (which chooses all the $k$ elements during the training phase–see (4) and the discussion therein). However, since Meta-Greedy does (a small) task-specific optimization at test time, its performance becomes much better. We emphasize again that, in order to be fair, the comparison in Figure 3 has been obtained using *the same* computational power allowed at test time for both meta-learning and two-stage approaches.

## Broader Impact

This paper introduces a discrete Meta-learning framework that aims at exploiting prior experience and data to improve performance on future tasks. While our results do not immediately lead to broader societal impacts, they can potentially lead to new approaches to reduce computation load and increase speed in latency-critical applications, such as recommender systems, autonomous systems, etc. In such applications, where there is a flow of new tasks arriving at any time requiring fast decisions, the available computational power and time is often limited either because we need to make quick decisions to respond to new users/tasks or since we need to save energy. For instance, in real-world advertising or recommendation systems, both these requirements are crucial: many users arrive within each hour which means fast decision-making is crucial, and also, reducing computation load would lead to huge energy savings in the long run. Our framework is precisely designed to address this challenge. Moreover, the framework introduced in this paper can potentially open new doors in the research field of discrete optimization.

## Acknowledgments and Disclosure of Funding

The research of Arman Adibi and Hamed Hassani is supported by NSF award CPS-1837253, NSF CAREER award CIF 1943064, and Air Force Office of Scientific Research Young Investigator Program (AFOSR-YIP) under award FA9550-20-1-0111. The research of Aryan Mokhtari is supported by NSF Award CCF-2007668.

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
