[Supplementary Material]

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

 optimal solution for problem (7). We first show that the output of algorithm 1 in phase 1 satisfies the following inequality:

$$\sum_{i=1}^m f_i(S_{\text{tr}}^* \cup S_i^*) - \sum_{i=1}^m f_i(S_{\text{tr}}) \le \sum_{i=1}^m f_i(S_{\text{tr}} \cup S_i^*) \tag{10}$$

To show (10) let $e^{(t)}$ be the $t^{th}$ element of greedy procedure in phase 1, and $S_{\text{tr}}^{(t)}$ be the $t^{th}$ set in this procedure, where $e^{(t)} = \arg\max_e \sum_{i=1}^m f_i(S_{\text{tr}}^{(t-1)} \cup e) - f_i(S_{\text{tr}}^{(t-1)})$. let $J^{(0)} = S_{\text{tr}}^*$ and define $J^{(t)}$ iteratively as follows. Let $D^{(t)} = J^{(t-1)} \setminus S_{\text{tr}}^{(t-1)}$ and define $o^{(t)}$ in the following way:

1.  If $e^{(t)} \in D^{(t)}$, then let $o^{(t)} = e^{(t)}$.

2.  Otherwise, if $e^{(t)} \notin D^{(t)}$, let $o^{(t)}$ be one of the elements of $D^t$ chosen uniformly at random.

Define $J^{(t)} := J^{(t-1)} \cup e^{(t)} \setminus o^{(t)}$. We show this procedure in the following chain.

$$(S_{\text{tr}}^*, \{S_i^*\}_{i=1}^m) \xrightarrow[\{o_i^{(1)}\}]{\{e_i^{(1)}\}} (J^{(1)}, \{S_i^*\}_{i=1}^m) \dots \xrightarrow[\{o_i^{(l)}\}]{\{e_i^{(l)}\}} (J^{(l)}, \{S_i^*\}_{i=1}^m)$$

$$(S_{\text{tr}} = \emptyset, \{S_i^0\}_{i=1}^m = \emptyset) \xrightarrow{\{e_i^{(1)}\}} (S_{\text{tr}}^{(1)}, \{\emptyset\}_{i=1}^m) \dots \xrightarrow{\{e_i^{(l)}\}} (S_{\text{tr}}^{(l)}, \{\emptyset\}_{i=1}^m)$$

then we can write the following inequalities:

$$\sum_{i=1}^m f_i(S_{\text{tr}}^{(t)}) - f_i(S_{\text{tr}}^{(t-1)}) = \sum_{i=1}^m f_i(S_{\text{tr}}^{(t-1)} \cup e^{(t)}) - f_i(S_{\text{tr}}^{(t-1)}) \tag{11}$$

$$\ge \sum_{i=1}^m f_i(S_{\text{tr}}^{(t-1)} \cup o_i^{(t)}) - f_i(S_{\text{tr}}^{(t-1)}) \tag{12}$$

$$\ge \sum_{i=1}^m f_i(S_i^* \cup J^{(t-1)}) - f_i(S_i^* \cup J^{(t-1)} \setminus o^{(t)}) \tag{13}$$

$$\ge \sum_{i=1}^m f_i(S_i^* \cup J^{(t-1)}) - f_i(S_i^* \cup J^{(t-1)} \setminus o_i^{(t)})$$

$$+ \sum_{i=1}^m -f_i(S_i^* \cup J^{(t)}) + f_i(S_i^* \cup J^{(t-1)} \setminus o_i^{(t)}) \tag{14}$$

$$= \sum_{i=1}^m f_i(S_i^* \cup J^{(t-1)}) - f_i(S_i^* \cup J^{(t)}) \tag{15}$$

where (12) follows from definition of $e^{(t)}$ and the greedy procedure and (13) follows from submodularity since in each step $S_{\text{tr}}^{(t-1)} \subseteq J^{(t-1)}$ and $o^{(t)} \notin S_{\text{tr}}^{(t-1)}$ and finally, equation (14) follows from the fact that $-f_i(J^{(t)} \cup S_i^*) + f_i(J^{(t-1)} \cup S_i^* \setminus o^{(t)}) \le 0$. Then, by summing over $t$ from 0 to $l$ we

get the following inequality:

$$\sum_{i=1}^{m} f_i(S_{\text{tr}}) = \sum_{i=1}^{m} f_i(S_{\text{tr}}^{(l)}) - f_i(S_{\text{tr}}^{(0)}) = \sum_{i=1}^{m}\sum_{t=0}^{l} f_i(S_{\text{tr}}^{(t)}) - f_i(S_{\text{tr}}^{(t-1)}) \tag{16}$$

$$\geq \sum_{i=1}^{m}\sum_{t=0}^{l} f_i(S_i^* \cup J^{(t-1)}) - f_i(S_i^* \cup J^{(t)}) \tag{17}$$

$$= \sum_{i=1}^{m} f_i(S_i^* \cup J^{(0)}) - f_i(S_i^* \cup J^{(l)}) \tag{18}$$

$$= \sum_{i=1}^{m} f_i(S_i^* \cup S_{\text{tr}}^*) - f_i(S_i^* \cup S_{\text{tr}}) \tag{19}$$

where the last equality comes from the process of defining $J$. Because, we only change one element by adding element found in greedy process and removing one element from the optimal set in each step and the size of $J^{(t)}$ is $l$ in each step; therefore, after $l$ step $J^{(l)} = S_{\text{tr}}$. By rearranging the terms and summing over $i$ the claim in (10) follows.

Second, for the phase 2 of the algorithm 1 we can use the usual analysis of greedy [30] for set $S_i$:

$$\sum_{i=1}^{m} f_i(S_{\text{tr}} \cup S_i) - f_i(S_{\text{tr}}) \geq (1 - \frac{1}{e})(\sum_{i=1}^{m} f_i(S_{\text{tr}} \cup S_i^{opt}) - f_i(S_{\text{tr}})) \tag{20}$$

$$\geq (1 - \frac{1}{e})(\sum_{i=1}^{m} f_i(S_{\text{tr}} \cup S_i^*) - f_i(S_{\text{tr}})) \tag{21}$$

$$\geq (1 - \frac{1}{e})(\sum_{i=1}^{m} f_i(S_{\text{tr}}^* \cup S_i^*) - 2f_i(S_{\text{tr}})) \tag{22}$$

where $S_i^{opt} = \arg\max_{|S_i| \leq k-l} f_i(S_{\text{tr}} \cup S_i)$ in the equation (20). Equation (20) follows from usual greedy analysis, equation (21) follows from definition of $S_{\text{tr}}^{opt}$, and equation (22) follows from equation (10).

Finally, since $S_i \subseteq S_i \cup S_{\text{tr}}$ by monotonicity $f_i(S_i \cup S_{\text{tr}}) \geq f_i(S_{\text{tr}})$. Then, combing this observation with the result in (22) implies

$$\sum_{i=1}^{m} f_i(S_{\text{tr}} \cup S_i) \geq \max\left\{\beta, \, (1 - 1/e)(\text{OPT} - 2\beta) + \beta\right\},$$

where $\beta := \sum_{i=1}^{m} f_i(S_{\text{tr}})$.

# 6    Proof of Proposition 2

Let $S_{\text{tr}}, \{S_i\}_{i=1}^{m}$ be the output of Algorithm 2 and $S_{\text{tr}}^*, \{S_i^*\}_{i=1}^{m}$ be the optimal solution for problem (7). We first show the following about the output of algorithm 2, phase 1.

$$\sum_{i=1}^{m} f_i(S_{\text{tr}}^* \cup S_i^*) - \sum_{i=1}^{m} f_i(S_i) \leq \sum_{i=1}^{m} f_i(S_{\text{tr}}^* \cup S_i) \tag{23}$$

to show (23) consider the following:

let $e_i^{(t)} = \arg\max_{e} f_i(S_i^{(t-1)} \cup e) - f_i(S_i^{(t-1)})$. let $J_i^{(0)} = S_i^*$ and define $J_i^{(t)}$ iteratively as follows. Let $D_i^t = J_i^{(t-1)} \setminus S_i^{(t-1)}$ and define $o_i^{(t)}$ in the following way:

1. If $e_i^{(t)} \in D_i^t$, then $o^{(t)} = e_i^{(t)}$;

2. Otherwise, if $e_i^{(t)} \notin D_i^t$, let $o_i^{(t)}$ be one of the elements of $D_i^t$ chosen uniformly at random;

Define $J_i^{(t)} := J_i^{(t-1)} \cup e_i^{(t)} \setminus o_i^{(t)}$.

$$(S_{\mathrm{tr}}^*, \{S_i^*\}_{i=1}^m) \xrightarrow[\{o_i^{(1)}\}]{\{e_i^{(1)}\}} (S_{\mathrm{tr}}^*, \{J_i^{(1)}\}_{i=1}^m) \ldots \xrightarrow[\{o_i^{(k-l)}\}]{\{e_i^{(k-l)}\}} (S_{\mathrm{tr}}^*, \{J_i^{(k-l)}\}_{i=1}^m)$$

$$(S_{\mathrm{tr}} = \emptyset, \{S_i^0\}_{i=1}^m = \emptyset) \xrightarrow{\{e_i^{(1)}\}} (\emptyset, \{S_i^{(1)}\}_{i=1}^m) \ldots \xrightarrow{\{e_i^{(k-l)}\}} (\emptyset, \{S_i^{(k-l)}\}_{i=1}^m)$$

then we can write the following inequalities:

$$f_i(S_i^{(t)}) - f_i(S_i^{(t-1)}) = f_i(S_i^{(t-1)} \cup e_i^{(t)}) - f_i(S_i^{(t-1)}) \tag{24}$$

$$\geq f_i(S_i^{(t-1)} \cup o_i^{(t)}) - f_i(S_i^{(t-1)}) \tag{25}$$

$$\geq f_i(S_{\mathrm{tr}}^* \cup J_i^{(t-1)}) - f_i(S_{\mathrm{tr}}^* \cup J_i^{(t-1)} \setminus o_i^{(t)}) \tag{26}$$

$$\geq f_i(S_{\mathrm{tr}}^* \cup J_i^{(t-1)}) - f_i(S_{\mathrm{tr}}^* \cup J_i^{(t-1)} \setminus o_i^{(t)})$$
$$- f_i(S_{\mathrm{tr}}^* \cup J_i^{(t)}) + f_i(S_{\mathrm{tr}}^* \cup J_i^{(t-1)} \setminus o_i^{(t)}) \tag{27}$$

$$= f_i(S_{\mathrm{tr}}^* \cup J_i^{(t-1)}) - f_i(S_{\mathrm{tr}}^* \cup J_i^{(t)}) \tag{28}$$

where (25) follows from definition of $e_i^{(t)}$ and the greedy procedure and (26) follows from the submodularity since in each step $S_i^{(t-1)} \subseteq J_i^{(t-1)}$ and $o_i^{(t)} \notin S_i^{(t-1)}$ and finally, equation (27) follows from the fact that $-f_i(S_{\mathrm{tr}}^* \cup J_i^{(t)}) + f_i(S_{\mathrm{tr}}^* \cup J_i^{(t-1)} \setminus o_i^{(t)}) \leq 0$ because of monotonicity. Then, by summing over $t$ from 0 to $k - l$ we get the following inequality:

$$f_i(S_i) = f_i(S_i^{(k-l)}) - f_i(S_i^{(0)}) = \sum_{t=0}^{k-l} f_i(S_i^{(t)}) - f_i(S_i^{(t-1)}) \tag{29}$$

$$\geq \sum_{t=0}^{k-l} f_i(S_{\mathrm{tr}}^* \cup J_i^{(t-1)}) - f_i(S_{\mathrm{tr}}^* \cup J_i^{(t)}) \tag{30}$$

$$= f_i(S_{\mathrm{tr}}^* \cup J_i^{(0)}) - f_i(S_{\mathrm{tr}}^* \cup J_i^{(k-l)}) \tag{31}$$

$$= f_i(S_{\mathrm{tr}}^* \cup S_i^*) - f_i(S_{\mathrm{tr}}^* \cup S_i) \tag{32}$$

where the last equality comes from the process of defining $J_i^{(k-l)}$; since, the size of $J_i^{(t)}$ is $k - l$ in each step and after $k - l$ step $J_i^{(k-l)} = S_i$. Then, by rearranging and summing over $i$ we can obtain (23).

Second, for phase 2 of algorithm 2 we can use the usual analysis of greedy [30] for set $S_{\mathrm{tr}}$ :

$$\sum_{i=1}^m f_i(S_{\mathrm{tr}} \cup S_i) - f_i(S_i) \geq (1 - \frac{1}{e})(\sum_{i=1}^m f_i(S_{\mathrm{tr}}^{opt} \cup S_i) - f_i(S_i)) \tag{33}$$

$$\geq (1 - \frac{1}{e})(\sum_{i=1}^m f_i(S_{\mathrm{tr}}^* \cup S_i) - f_i(S_i)) \tag{34}$$

$$\geq (1 - \frac{1}{e})(\sum_{i=1}^m f_i(S_{\mathrm{tr}}^* \cup S_i^*) - 2f_i(S_i)) \tag{35}$$

where $S_{\mathrm{tr}}^{opt} = \underset{|S_{\mathrm{tr}}| \leq l}{\arg\max} \sum_{i=1}^m f_i(S_{\mathrm{tr}} \cup S_i)$ in equation (33). Equation (33) follows from the usual greedy analysis, equation (34) follows from the definition of $S_{\mathrm{tr}}^{opt}$, and equation (35) follows the from equation (23).

Finally, since $S_i \subseteq S_i \cup S_{\text{tr}}$ by monotonicity $f_i(S_i \cup S_{\text{tr}}) \geq f_i(S_i)$. Then, combing (33)-(35) we have:

$$\sum_{i=1}^{m} f_i(S_{\text{tr}} \cup S_i) \geq \max\left\{\gamma, (1 - 1/e)(\text{OPT} - 2\gamma) + \gamma\right\}.$$

The following shows the ratio of lower bound to optimum (a similar plot can be obtained for the lower bound of Proposition 1 when $\gamma$ is replaced with $\beta$.).

Figure 4: y-axis: The lower bound of Proposition 2 divided by OPT, x-axis: $\gamma/\text{OPT}$.

## 7 Proof of Theorem 1

Let $\theta_2 = \sum_{i=1}^{m} f_i(S_{\text{tr}}^{(2)} \cup S_i^{(2)})$. Since $S_{\text{tr}}^{(2)}$ found greedily given $\{S_i\}_{i=1}^{m}$ we can write:

$$\theta_2 - \gamma \geq (\text{OPT} - \gamma)(1 - \frac{1}{e}) \geq (\sum_{i=1}^{m} f_i(S' \cup S_i^{(2)}) - \gamma)(1 - \frac{1}{e}) \tag{36}$$

for every $\mid S' \mid \leq l$. Also, we can write

$$\text{OPT} - \gamma = \sum_{i=1}^{m} f_i(S_{\text{tr}}^* \cup S_i^*) - f_i(S_i^{(2)}) \tag{37}$$

$$\leq \sum_{i=1}^{m} f_i(S_{\text{tr}}^* \cup S_i^{(2)} \cup S_i^*) - f_i(S_i^{(2)}) \tag{38}$$

$$= \sum_{i=1}^{m} f_i(S_{\text{tr}}^* \cup S_i^{(2)} \cup S_i^*) + f_i(S_{\text{tr}}^* \cup S_i^{(2)}) - f_i(S_{\text{tr}}^* \cup S_i^{(2)}) - f_i(S_i^{(2)}) \tag{39}$$

$$\leq \sum_{i=1}^{m} f_i(S_{\text{tr}}^* \cup S_i^{(2)} \cup S_i^*) - f_i(S_{\text{tr}}^* \cup S_i^{(2)}) + \frac{\theta_2 - \gamma}{1 - 1/e} \tag{40}$$

$$\leq \sum_{i=1}^{m} f_i(S_i^{(2)} \cup S_i^*) - f_i(S_i^{(2)}) + \frac{\theta_2 - \gamma}{1 - 1/e} \tag{41}$$

where (40) comes from (36), and (41) comes from submodularity. We thus obtain

$$\text{OPT} - \frac{\theta_2 - \gamma}{1 - 1/e} - \gamma \leq \sum_{i=1}^{m} f_i(S_i^{(2)} \cup S_i^*) - f_i(S_i^{(2)}) \tag{42}$$

Also we can write for any set $S'$ such that $|S'| \leq l$:

$$\sum_{i=1}^{m} f_i(S' \cup S_i^*) - f_i(S') \geq \sum_{i=1}^{m} f_i(S' \cup S_i^* \cup S_i) - f_i(S' \cup S_i) \tag{43}$$

$$\geq \sum_{i=1}^{m} f_i(S' \cup S_i^* \cup S_i) - f_i(S_i) + f_i(S_i) - f_i(S' \cup S_i) \tag{44}$$

$$\geq \sum_{i=1}^{m} f_i(S_i^* \cup S_i) - f_i(S_i) + f_i(S_i) - f_i(S' \cup S_i) \tag{45}$$

$$\geq \text{OPT} - \frac{\theta_2 - \gamma}{1 - 1/e} - \gamma + \sum_{i=1}^{m} f_i(S_i) - f_i(S' \cup S_i) \tag{46}$$

$$\geq \text{OPT} - 2\frac{\theta_2 - \gamma}{1 - 1/e} - \gamma \tag{47}$$

where (43) follows from submodularity, (45) follows from monotonicity, and (46) follows from (42), and (47) follows from (36). This results the following for any set $S'$ such that $|S'| \leq l$:

$$\sum_{i=1}^{m} f_i(S' \cup S_i^*) - f_i(S') \geq \text{OPT} - 2\frac{\theta_2 - \gamma}{1 - 1/e} - \gamma \tag{48}$$

Now, from (48) we can find a new bound for the performance of algorithm 3. From (48) we can write:

$$\sum_{i=1}^{m} f_i(S_{\text{tr}}^{(1)} \cup S_i^*) - f_i(S_{\text{tr}}^{(1)}) \geq \text{OPT} - 2\frac{\theta_2 - \gamma}{1 - 1/e} - \gamma \tag{49}$$

Also, since in Algorithm 1 the set $S_i^{(1)}$ is constructed greedily on the top of $S_{\text{tr}}^{(1)}$, we have:

$$\sum_{i=1}^{m} f_i(S_{\text{tr}}^{(1)} \cup S_i^{(1)}) - \beta \geq (\sum_{i=1}^{m} f_i(S_{\text{tr}}^{(1)} \cup S_i^*) - \beta)(1 - \frac{1}{e}) \tag{50}$$

$$\geq (\text{OPT} - 2\frac{\theta_2 - \gamma}{1 - 1/e} - \gamma)(1 - \frac{1}{e}), \tag{51}$$

where (51) follows from (49). We thus obtain:

$$\sum_{i=1}^{m} f_i(S_{\text{tr}}^{(1)} \cup S_i^{(1)}) \geq (\text{OPT} - 2\frac{\theta_2 - \gamma}{1 - 1/e} - \gamma)(1 - \frac{1}{e}) + \beta \tag{52}$$

Using the same procedure as above, by defining $\theta_1 = \sum_{i=1}^{m} f_i(S_{\text{tr}}^{(1)} \cup S_i^{(1)})$, we can prove:

$$\sum_{i=1}^{m} f_i(S_{\text{tr}}^{(2)} \cup S_i^{(2)}) \geq (\text{OPT} - 2\frac{\theta_1 - \gamma}{1 - 1/e} - \gamma)(1 - \frac{1}{e}) + \beta \tag{53}$$

which results in the following lower bound:

$$\max\left\{\sum_{i=1}^{m} f_i(S_{\text{tr}}^{(1)} \cup S_i^{(1)}), \sum_{i=1}^{m} f_i(S_{\text{tr}}^{(2)} \cup S_i^{(2)})\right\}$$

$$\geq \max\left\{\theta_1, \theta_2, (1 - 1/e)(\text{OPT} - \gamma) + \beta - 2(\theta_2 - \gamma), (1 - 1/e)(\text{OPT} - \beta) + \gamma - 2(\theta_1 - \beta)\right\}. \tag{54}$$

Finally, given (52) and (54), the factor 0.53 is obtained as a result of the following procedure. Let $\beta$ and $\gamma$ given as $\beta := \sum_{i=1}^{m} f_i(S_{\text{tr}}^{(1)})$ and $\gamma := \sum_{i=1}^{m} f_i(S_i^{(2)})$. Then the left-hand-side term in (8) is

lower bounded by:

$$\min_{\theta_1,\theta_2} \; \max\left\{\theta_1, \theta_2, (1-1/e)(\text{OPT}-\gamma) + \beta - 2(\theta_2 - \gamma), (1-1/e)(\text{OPT}-\beta) + \gamma - 2(\theta_1 - \beta)\right\}$$
$$\text{subject to} \quad \theta_1 \geq \max\{\beta, (1-1/e)(\text{OPT} - 2\beta) + \beta\}$$
$$\theta_2 \geq \max\{\gamma, (1-1/e)(\text{OPT} - 2\gamma) + \gamma\}$$

Note that the constraints hold due to the results of Proposition 1 and 2. In particular, the above bound is always larger than $0.53 \times \text{OPT}$ for any value of $\beta$ and $\gamma$.

# 8 Proof of Theorem 2

Consider round $t$ in which $\mid S_{\text{tr}} \mid < l$ and $\mid S_i \mid < k - l$ the expected gain of the algorithm with probability $\frac{l}{k}$ is the maximum gain from adding an element $e^* = \arg\max_e \sum_{i=1}^{m} f_i(S_{\text{tr}}^t \cup e \cup S_i^t) - f_i(S_{\text{tr}}^t \cup S_i^t)$ or with probability $\frac{k-l}{k}$ the gain is $\sum_{i=1}^{m} \max_{e_i} f_i(S_{\text{tr}}^t \cup e_i \cup S_i^t) - f_i(S_{\text{tr}}^t \cup S_i^t)$ which can be written as follows.

$$\mathbb{E}[\sum_{i=1}^{m} f_i(S_{\text{tr}}^{t+1} \cup S_i^{t+1}) - f_i(S_{\text{tr}}^t \cup S_i^t)|S_{\text{tr}}^t, S_i^t]$$
$$= \frac{l}{k} \max_e \sum_{i=1}^{m} f_i(S_{\text{tr}}^t \cup e \cup S_i^t) - f_i(S_{\text{tr}}^t \cup S_i^t) + \frac{k-l}{k} \sum_{i=1}^{m} \max_{e_i} f_i(S_{\text{tr}}^t \cup e_i \cup S_i^t) - f_i(S_{\text{tr}}^t \cup S_i^t) \tag{56}$$

assuming $S_{\text{tr}}^*, S_i^*$ is optimal solution, we can also write:

$$\frac{1}{k} \sum_{i=1}^{m} f_i(S_{\text{tr}}^* \cup S_i^*) - f_i(S_{\text{tr}}^t \cup S_i^t) \leq \frac{1}{k} \sum_{i=1}^{m} f_i(S_{\text{tr}}^* \cup S_i^* \cup S_{\text{tr}}^t \cup S_i^t) - f_i(S_{\text{tr}}^t \cup S_i^t) \tag{57}$$

$$\leq \frac{1}{k} \sum_{e \in S_{\text{tr}}^* \backslash S_{\text{tr}}^t} \sum_{i=1}^{m} f_i(e \cup S_{\text{tr}}^t \cup S_i^t) - f_i(S_{\text{tr}}^t \cup S_i^t)$$

$$+ \frac{1}{k} \sum_{i=1}^{m} \sum_{e \in S_i^* \backslash S_i^t} f_i(e \cup S_{\text{tr}}^t \cup S_i^t) - f_i(S_{\text{tr}}^t \cup S_i^t) \tag{58}$$

$$\leq \frac{l}{k} \max_e \sum_{i=1}^{m} f_i(S_{\text{tr}}^t \cup e \cup S_i^t) - f_i(S_{\text{tr}}^t \cup S_i^t)$$

$$+ \frac{k-l}{k} \sum_{i=1}^{m} \max_{e_i} f_i(S_{\text{tr}}^t \cup e_i \cup S_i^t) - f_i(S_{\text{tr}}^t \cup S_i^t) \tag{59}$$

where (57) follows from monotonicity, and (58) follows from submodularity. Then, from (59) and (56) we conclude that:

$$\mathbb{E}[\sum_{i=1}^{m} f_i(S_{\text{tr}}^{t+1} \cup S_i^{t+1}) - f_i(S_{\text{tr}}^t \cup S_i^t)|S_{\text{tr}}^t, S_i^t] \leq \frac{1}{k} \sum_{i=1}^{m} f_i(S_{\text{tr}}^* \cup S_i^*) - f_i(S_{\text{tr}}^t \cup S_i^t) \tag{60}$$

In other words, the expected improvement in the objective (left-hand side of (60)) is at least $1/k$ times the gap of the current objective value to OPT (i.e. right-hand side of (60)). Note that (60) is only valid when $\mid S_{\text{tr}} \mid < l$ and $\mid S_i \mid < k - l$. Hence, by defining the stopping time $\tau$ as first time that either $\mid S_{\text{tr}} \mid = l$ or $\mid S_i \mid = k - l$, and a telescopic usages of the bounds in (60), we obtain the following bound:

$$\mathbb{E}[\sum_{i=1}^{m} f_i(S_{\text{tr}}^{\tau} \cup S_i^{\tau})] \geq \text{OPT} \, \mathbb{E}[(1 - (1 - \frac{1}{k})^{\tau})]$$

The following theorem finds an upper bound on $\mathbb{E}[(1 - \frac{1}{k})^{\tau}]$ which finishes the proof.

**Lemma 1.** *If stopping time $\tau$ is first time that either $\mid S_{\text{tr}} \mid = l$ or $\mid S_i \mid = k - l$ then $\mathbb{E}[(1 - \frac{1}{k})^\tau] \leq c + \exp(-1 + \sqrt{3c.log(\frac{k}{c})})$ where $c = \frac{1}{\min\{l, k-l\}}$.*

*Proof.* let $u_1, u_2, \cdots$ be *i.i.d* random variables with distribution $u_i \sim$ Bernoulli$((k - l)/k)$, i.e. $p(u_i = 1) = (k-l)/k$. The stopping time $\tau$ is the first time that $\sum_{i=1}^\tau u_i = k - l$ or $\tau - \sum_{i=1}^\tau u_i = l$. Let us define $X_r = \sum_{i=1}^r u_i$.

Furthermore, we define $\tau' = r$ when $r$ is the first time that $X_r = r - l$ and $\tau'' = r$ when $r$ is the first time that $X_r = k - l$. Also, let $c = \frac{1}{\min\{l, k-l\}}$ as it was defined in the lemma. By this definition, $\tau = \min\{\tau'', \tau'\}$ and we can write the following about the probabilities of $\tau'$ and $\tau''$:

$$p(\tau' = r) = \binom{r - 1}{l - 1}(\frac{k - l}{k})^{r-l}(\frac{l}{k})^l$$

$$p(\tau'' = r) = \binom{r - 1}{k - l - 1}(\frac{l}{k})^{r-k+l}(\frac{k - l}{k})^{k-l}$$

then, based on the definition of $\tau'$ and $\tau''$ we have the following properties for $\tau'$ and $\tau''$:

- if $r < k - l$ then $p(\tau'' = r) = 0$.

- if $r < l$ then $p(\tau' = r) = 0$.

- if $r > k$ then $p(\tau' \leq \tau'' | \tau' = r) = 0$.

- if $r < k$ then $p(\tau' \leq \tau'' | \tau' = r) = 1$.

- if $r < k$ then $p(\tau' \geq \tau'' | \tau'' = r) = 1$

- if $r > k$ then $p(\tau' \geq \tau'' | \tau'' = r) = 0$.

- $p(\tau'' = r | \tau' \geq \tau'') = p(\tau = r | \tau' \geq \tau'')$.

- $p(\tau' = r | \tau' \leq \tau'') = p(\tau = r | \tau' \leq \tau'')$.

Moreover using Bayes rule we can write:

- 
$$p(\tau' = r | \tau' \leq \tau'') = \frac{p(\tau' \leq \tau'' | \tau' = r)p(\tau' = r)}{p(\tau' \leq \tau'')} = \frac{\mathbb{1}(r \leq k)p(\tau' = r)}{p(\tau' \leq \tau'')}.$$

- 
$$p(\tau'' = r | \tau' \geq \tau'') = \frac{\mathbb{1}(r \leq k)p(\tau'' = r)}{p(\tau'' \leq \tau')}.$$

Let $\bar{X}_r = r - X_r$ we can write $\bar{X}_r = \sum_{i=1}^r v_i$ where $v_1, v_2, v_3, \ldots$ are *i.i.d* random variable with distribution $v_i \sim$ Bernoulli$((l)/k)$. Then, we can write the following using Chernoff bound:

$$p(\tau' = r) \leq p(X_r = r - l) \tag{61}$$

$$\leq p(\bar{X}_r \geq l) \tag{62}$$

$$\leq p(\bar{X}_r \geq r(\frac{l}{k}) - (k - r)\frac{l}{k}) \tag{63}$$

$$\leq \exp\left(-\frac{(k - r)^2(\frac{l}{k})^2}{3r(\frac{l}{k})}\right) \tag{64}$$

$$= \exp\left(-\frac{(k - r)^2(l)}{3rk}\right) \tag{65}$$

Similarly:

$$p(\tau'' = r) \le p(X_r = k - l) \tag{66}$$

$$\le p(X_r \ge k - l) \tag{67}$$

$$\le p(X_r \ge r(1 - \frac{l}{k}) - (k - r)(1 - \frac{l}{k})) \tag{68}$$

$$\le \exp\left(-\frac{(k-r)^2(1-\frac{l}{k})^2}{3r(1-\frac{l}{k})}\right) \tag{69}$$

$$\le \exp\left(-\frac{(k-r)^2(k-l)}{3rk}\right) \tag{70}$$

$$\le \exp\left(-\frac{(k-r)^2}{3rkc}\right) \tag{71}$$

then we can write the $\mathbb{E}[(1 - \frac{1}{k})^\tau]$ as follows:

$$\mathbb{E}[(1 - \frac{1}{k})^\tau] = \sum_{r=1}^{k}(1 - \frac{1}{k})^r p(\tau = r) \le (1 - \frac{1}{k})^{k-\alpha\sqrt{c}} + \sum_{r=1}^{k-\alpha\sqrt{c}}(1 - \frac{1}{k})^r p(\tau = r) \tag{72}$$

Our goal is to find proper bound for (72). we focus on the second term in (73)-(79) and try to find proper bound for it.

$$\sum_{r=1}^{k-\alpha\sqrt{c}}(1 - \frac{1}{k})^r p(\tau = r) \tag{73}$$

$$= \sum_{r=1}^{k-\alpha\sqrt{c}}(1 - \frac{1}{k})^r (p(\tau' = r|\tau' < \tau'')p(\tau' < \tau'') + p(\tau'' = r|\tau' \ge \tau'')p(\tau' \ge \tau'')) \tag{74}$$

$$= \sum_{r=1}^{k-\alpha\sqrt{c}}(1 - \frac{1}{k})^r (p(\tau' = r) + p(\tau'' = r)) \tag{75}$$

$$= \sum_{r=l}^{k-\alpha\sqrt{c}}(1 - \frac{1}{k})^r p(\tau' = r) + \sum_{r=k-l}^{k-\alpha\sqrt{c}}(1 - \frac{1}{k})^r p(\tau'' = r) \tag{76}$$

$$\le \sum_{r=l}^{k-\alpha\sqrt{c}}\exp\left(-\frac{(k-r)^2}{3rkc}\right) + \sum_{r=k-l}^{k-\alpha\sqrt{c}}\exp\left(-\frac{(k-r)^2}{3rkc}\right) \tag{77}$$

$$\le (k-l)\exp\left(-\frac{(k-(k-\alpha\sqrt{c}))^2}{3k^2c}\right) + l\exp\left(-\frac{(k-(k-\alpha\sqrt{c}))^2 l}{3k^2}\right) \tag{78}$$

$$\le (k-l)\exp\left(-\frac{(\alpha\sqrt{c})^2}{3k^2c}\right) + l\exp\left(-\frac{(\alpha\sqrt{c})^2 l}{3k^2}\right) \tag{79}$$

where (74) follows from law of total probability, (75) follows from bayes rule, (77) follows from Chernoff bound, (78) follows from the fact that $r < k$. Let $\alpha = 3\sqrt{\log(\frac{1}{c})}.k$. As result, we have:

$$\sum_{r=1}^{k-\alpha\sqrt{c}}(1 - \frac{1}{k})^r p(\tau = r) \le (k-l)c^3 + lc^{3cl} \tag{80}$$

Assume without loss of generality $k - l \le l$ and $k - l \ge 2$. As a result, $c = \frac{1}{k-l}$. we want to show that $(k-l)c^3 + lc^{3cl} = c^2 + lc^{3cl} \le c$. To show this, we show the following equivalent inequality :

$$l(k-l)^{-3cl} \le c(1-c) = \frac{k-l-1}{(k-l)^2} \tag{81}$$

This holds since $k - l \geq 2$ we have $\frac{l}{(k-l)^3}(k-l)^{-3(cl-1)} \leq \frac{l}{(k-l)^3}2^{-3(\frac{l}{k-l}-1)} \leq \frac{l}{(k-l)^3}\frac{l}{k-l} = \frac{1}{(k-l)^2} \leq \frac{k-l-1}{(k-l)^2}$. Moreover, we can bound the first term in (72) as follows:

$$(1 - \frac{1}{k})^{k-\alpha\sqrt{c}} \leq \exp(-1 + 3\sqrt{c.\log(\frac{1}{c})}) \tag{82}$$

summing up we can find the following bound for $\mathbb{E}[(1 - \frac{1}{k})^\tau]$ which finishes the proof.

$$\mathbb{E}[(1 - \frac{1}{k})^\tau] \leq c + \exp(-1 + 3\sqrt{c.log(\frac{1}{c})}) \tag{83}$$

∎

## 9 Counter-example for Submodularity of the Objective in (7)

In this section, we provide a counterexample for submodularity of the objective function in the equation (7). We consider a maximum coverage problem in which the function value is an area covered by a set of elements. We define the ground set $V = \{ABIJ, BCDI, ACDJ, IDEH, HEFG, BCEH\}$ which has shown in Figure 5. Each element is a rectangle, and a function value of that element is an area covered by that element. We refer to each element (rectangle) by it's vertices.

Figure 5: Counter Example of Submodularity

Let $AC = CD = DE = EF = 1$, and $BC = 0.75$. Also in (7) we let $m = 1$ and $k - l = 1$ which means that we are considering a single set function $f$ defined as: $f(S) = \max_{e \in V} A(S \cup e)$, where $A(T)$ is a area of set $T$. Note that the area function $A$ is monotone and submodular, however as we will show below, the function $f$ is not submodular. To do so, we consider two sets $T_1 = \emptyset$ and $T_2 = \{ACDJ\}$ and add the element $IDEH$ to both sets and observe that $f$ does not satisfy the diminishing returns property. Let us first compute the function value at $T_1$ and $T_2$ as follows:

$$f(T_1) = \max_{e \in V} A(e) = A(\{BCEH\}) = 1.5,$$

and

$$f(T_2) = \max_{e \in V} A(T_2 \cup e) = A(\{ACDJ, IDEH\}) = 1.75.$$

Similarly, we compute the function value at $T_1' = T_1 \cup \{IDEH\}$, and $T_2' = T_2 \cup \{IDEH\}$:

$$f(T_1') = \max_{e \in V} A(T_1' \cup e) = A(\{IDEH, ACDJ\}) = 1.75,$$

and
$$f(T_2^{'}) = \max_{e \in V} A(T_2^{'} \cup e) = A(\{IDEH, ACDJ, EFGH\}) = 2.5.$$

We can now see that $T_1 \subseteq T_2$, but $f(T_2^{'}) - f(T_2) \not\preceq f(T_1^{'}) - f(T_1)$. Therefore, $f$ is not submodular.

Also let us make a remark about $k$-submodularity which studies functions of $k$ subsets of the ground set that are disjoint sets. This class of functions is submodular in each orthant [48]. However, in the submodular meta-learning framework, sets can have overlap, and there is no restriction on the sets to be disjoint. Therefore, our framework is different from $k$-submodular maximization.