[Reviews · NeurIPS 2020]

Review 1

Summary and Contributions: The main contribution of this work is an algorithm for meta-learning in the setting where the objective function is a submodular set function. Given a sample of functions as training data, the aim is to precompute a set S_tr such that for any test function drawn from the same distribution, we can create a near-optimal set by augmenting S-tr by a small number of elements.

Strengths: The problem seems well motivated, the results are theoretically sound. The authors present two algorithms, as well as a meta algorithm that combines the two. From the sketch given in the supplementarya, the proofs appear quite non-trivial and novel. They further present a randomized algorithm with better guarantees. I particularly liked the intuition that combining the two algorithms provides a better guarantee.

Weaknesses: Nothing in particular. My initial reaction was that creating m different S_i's is very expensive. However, this seems to be the first algorithm for this problem, so this is not really a weakness.

Correctness: I scanned through part of the supplementary proofs, they look correct.

Clarity: There are some minor typos that I note below. Other than that the paper is well written.

Relation to Prior Work: Prior work is this setting has been limited to continuous functions. The current work claims to be the first one to extend it to set functions.

Reproducibility: Yes

Additional Feedback: ---- Post Rebuttal comments: After reading the author rebuttal, I am happy to maintain my score. ---- In Algo 1 is written in a somewhat confusing manner. The text description is clearer and in saying that each S_i is constructed, for all possible i, this is not evident from the formal algorithm description (since line .. is outside of the for-loop and hence i is a free variable there). Similarly, Meta-Greedy doesn’t just output S_tr and a single S_i, it should be {S_i, i = 1..m} Thanks for remark 1, it might do it good to push it up earlier, else it is not clear why the algorithms proposed are returning a number of sets {S_i}.


Review 2

Summary and Contributions: The authors propose a preprocessing approach to speed up submodular maximization query. Specifically, given a distribution over submodular functions and cardinality budget of $k$, the authors propose to pick $\ell < k$ elements that maximize the average submodular function, and only compute the remaining $k-\ell$ separately for each function. They show that a simple greedy approach that at each iteration adds either one of the $\ell$ or one of the $k-\ell$ elements obtains the usual $1-1/e$ approximation, if done at random. They also demonstrate in experiments that the combination of preprocesss- and query-phase elements can give good average values for natural distributions.

Strengths: The experiments suggest that this is a promising approach that may be helpful in practice. (Although TBH with only 2 experiments it’s hard to tell how well it would perform on other tasks.)

Weaknesses: The main theoretical result (for randomized algorithm) follows by standard argument so the contribution is relatively thin. The algorithm is also not very surprising. I think that the authors missed the point of Broader Impact statement. They advocate optimizing a one-size-almost-fits-all solution that is good *on average*, but will perform poorly for some functions (e.g. Netflix users). Instead of discussing this issue, they make some generic statement how slightly faster algorithms are better for saving energy.

Correctness: Yes (I did not verify Theorem 1)

Clarity: Overall pretty good, but the writing in the intro is a bit repetitive. The space would have been better used giving a hint to proof of Theorem 2 or even the counter example to submodularity.

Relation to Prior Work: Yes

Reproducibility: Yes

Additional Feedback:


Review 3

Summary and Contributions: This paper introduces the problem of submodular meta-learning, which is a discrete variant of the problem of Meta-learning in the continuous domain. The goal is to use data from previous tasks to initialize a solution that can then be adapted for a new task using a small number of queries. Here, each of the tasks corresponds to a submodular function, an initial solution corresponds to a collection of k - l elements, and a final solution for a new task corresponds to choosing l elements to complete the initial solution. The authors give deterministic and randomized algorithms that obtain a .53 and close to 1-1/e approximations for this problem. Post rebuttal: I appreciate the thorough answers to my comments in the rebuttal and would like to keep my strong score for this paper.

Strengths: - An interesting and well-motivated new problem. Meta-learning is an important problem that has received a lot of attention in the continuous domain. The authors propose a natural meta-learning framework for discrete optimization. The application to the problem of recommending a set of items is strong. - Strong results for a non-trivial problem. The problem is non-submodular and constant factor approximation are obtained. It's also interesting that combining Algorithm 1 and Algorithm 2 does better than the algorithms individually. - The paper is very well-written

Weaknesses: No major weaknesses. Minor weakness: - For the computation cost at test-time to be significantly reduced compared to running the full greedy algorithm, the number k-l of elements that still need to be chosen must be very small, which doesn't leave room for a lot of personalization in the example of item recommendation. - The authors show a better approximation by combining algorithms 1 and 2, but is it the case that the same approximation as for alg 3 could potentially be obtained for algorithm 1 or 2? If no, it would be nice to include in the appendix some example where alg 1 and 2 don't obtain better than a 1/2 approximation.

Correctness: I did not find any errors.

Clarity: Yes, very well.

Relation to Prior Work: Yes, the relation to prior work is clearly discussed. Minor comment: the authors claim that "the main difference of our framework with the two-stage approaches is that we allow for personalization". This claim seems too strong, the two-stage framework also allows for personalization.

Reproducibility: Yes

Additional Feedback: In addition to first choosing l elements at train time and then k-l elements at test time, another approach could be to choose k elements at train time and then allow some number of swaps between these k elements and the remaining elements. This could allow for additional personalization as it would avoid having the same l elements recommended to all users? Minor comment: -line 86, "l" does not appear in equation (1).


Review 4

Summary and Contributions: In this paper, the authors describe a meta-learning framework extension to the discrete setting. Specifically, they consider the cases where the functions that the tasks aim to maximize are monotone and submodular set functions. The authors first present two deterministic greedy algorithms, one which constructs S_tr first and another which constructs the sets at test time first. They show that both of these algorithms are at least 1/2-optimal. They also present a meta-greedy algorithm (which chooses the better solution between the previous two algorithms) and prove that it is at least 0.53-optimal. The authors also present a randomized meta-greedy algorithm. Finally, the paper also describes results from applying the above algorithms to two practical problems: ride-sharing and movie-recommendation.

Strengths: The authors did a great job. They have proposed a few techniques for a relevant problem faced by the ML community. They substantiated their proposed techniques with proofs (mostly in the supplement, which I was able to skim through)—I did not find any immediate flags. I appreciated that the experiments used baselines and an oracle to provide the reader with more context for how well their algorithms perform (with the possible caveat I note in the next section). This paper is not in my area of expertise and so I do not have much to contribute to this review.

Weaknesses: The greedy-test seems to be the oracle and the other benchmarks (greedy-train and random) seem to be baselines. I’m curious how the proposed algorithm compares to other approaches the authors mentioned in the related work section. I understand that these approaches are not concerned with train-test optimization, but it may be helpful to compare those techniques to the the proposed one in terms of optimality. There seems to be a minor error on line 86: “l” isn’t defined. I did not identify any significant weakness. Please keep in mind that this is not in the area of my expertise.

Correctness: While I was able to follow the paper, understand the authors’ claims, and not find any obvious errors, I do not have expertise in this particular area and thus, I will refrain from commenting on the correctness of the claims in this paper.

Clarity: Yes, I thank the authors for this. I especially appreciated that they grounded the problem setup and their proposed solution with the example of recommendation systems.

Relation to Prior Work: Yes, the paper listed several recent works and substantiated how their work differed from and built upon those works. However, given my lack of expertise in this particular domain, I’m unable to comment on the comprehensiveness of their literature review.

Reproducibility: Yes

Additional Feedback:

[Author Response · NeurIPS 2020]

We thank the reviewers for their careful consideration and constructive feedback. Below, please find our responses.

**General comments. GC1:** In order to highlight the novelty of our theoretical results, we'd like to emphasize that the
meta-learning objective considered in this paper in *not* submodular and hence providing constant-factor approximations
is indeed novel and non-trivial. In the paper, we have provided two different methods: (1) The Meta-Greedy algorithm
(Alg. 3) which combines two carefully-chosen deterministic orderings for greedy-selection and leads to a solution
that is at least $0.53$-optimal (which is indeed not a common result). The proof of Theorem 1 (and Propositions 1,2)
introduces new techniques to carefully analyze the interplay between the inner and outer maximization problems in
the meta-learning objective. (2) We show that by selecting elements according to a *properly-designed* randomized
procedure we obtain in expectation an $1 - 1/e - o(1)$-optimal solution, where the $o(1)$ term vanishes when $l$ and $k - 1$
grow (see Theorem 2 and the probabilistic analysis therein). This result is also novel and non-trivial, since $1 - 1/e$ is
only common for submodular and monotone problems, while our problem is not submodular. Further, besides all the
theoretical contributions, another main novelty of this paper is to introduce the first discrete meta-learning framework.

**GC2:** We'd like to emphasize that we have compared our proposed scheme with one of
the main schemes for two-stage submodular optimization (i.e. the Replacement-Greedy
method). The comparison and the plot was provided in the supplementary materials,
however, in the revised version we will include it in the main body. As illustrated in
the figure on the right, our proposed methods lead to a better user-specific solution (on
average) compared to the two-stage algorithm (Replacement-Greedy). For the details
of this experiment please check the supplementary material of the submitted paper.

**Reviewer #1. Q1:** The cost of creating $m$ different $S_i$'s. **A1:** Thanks for this careful
comment. Please note that in Algorithm 1 these sets can be computed in parallel (as their selection process are
independent) to improve the run-time of the algorithm to $kn$. We'll highlight this point in the final submission. Given
that $m$ tasks are involved, any good algorithm would have a complexity dependent on $m$, however, parallelization could
significantly reduce the run-time. **Q2:** Clarity in Alg. 1 and bringing Remark 1 up. **A2:** Thanks for your suggestion:
We'll modify Alg. 1 to clarify that $m$ sets $S_i$ are constructed for all possible $i = 1, \ldots, m$. We'll also clarify that
Meta-Greedy outputs $m + 1$ sets, i.e., $S_{tr}$ and $\{S_i\}_{i=1}^m$, and state Remark 1 earlier after Alg. 1.

**Reviewer #2. Q3:** Importance and novelty of the theoretical results. **A3:** Please read our general comment (GC1)
above. **Q4:** The point of Broader Impact. **A4:** Regarding the broader impact section, we appreciate your comment
and will revise this section by better highlighting the fact that our proposed scheme leads to personalized solutions for
different users instead of providing a single solution that might one average work well but could perform poorly for
some users (e.g. Netflix users). **Q5:** Better use of space **A5:** Following your suggestions, we will include the counter
example to submodularity in the main body and will also provide a hint to the proof for Theorems 1,2. These changes
will be possible by shortening the introduction and using the extra page (9th page) for the final submission. Thank you.

**Reviewer #3. Q6:** Same guarantee of Alg. 3 for Algs. 1,2? **A6:** Thank you for this great and insightful question. Alg.
3 performs *strictly* better than each of Alg. 1 and Alg. 2. Indeed, one can provide simple examples where Alg. 1 or Alg.
2 (either of them, not both) obtains a solution which is exactly 1/2-optimal. However, the best solution among Alg. 1
and Alg. 2 will be at least $0.53$-optimal. Also, the best performing algorithm among Alg. 1 and Alg. 2 could change for
different problems, and this is why we use the maximum of them as the output of our Meta-Greedy method (Alg. 3). We
will include some simple examples to better highlight this point. **Q7:** Two-stage approaches and personalization. **A7:**
The reviewer is absolutely right that two-stage also allows for personalization. We'll highlight this point and revise the
wording of the sentence that the reviewer has mentioned. The main difference is in the two stage approach personalized
items could be selected only from a smaller ground set (which is tailored to the training tasks), while in our approach
those elements can be selected from the original ground set. Due to this difference our approach could possibly lead to
a better personalized solution as illustrated in the figure above (and general comment GC2). **Q8:** Limited computation
at test time and room for personalization. **A8:** As the reviewer has correctly pointed out, the proposed approach will be
highly beneficial when the number of elements added at test time (i.e., $k - l$) is much smaller than the size of solution
set (i.e., $k$). Please note that this is aligned with the main point of meta-learning which is to find a solution at training
time that can be quickly adapted at test time (with very low computation) to the new ask. Indeed, this assumption is
realistic, as in most cases, the solutions for different users have several common elements, and only differ on a small
number of elements. We hope that the examples in the paper and the experiments could convey this point. **Q9:** Idea:
Swapping at test time. **A9:** The reviewer's idea to swap the elements at test time instead of adding elements is very
interesting, and it could be a future direction to explore. We'll include it in our concluding remarks.

**Reviewer #4.** Thanks for your insightful suggestions and careful reading. **Q10:** Comparison with relevant work. **A10:**
The closest framework to our setting is the two-stage submodular optimization framework. We have actually compared
our proposed Meta-Learning approach with a state-of-the-art two-stage scheme. The comparison (and plot) is provided
in the supplementary materials. Please see the general comment (GC2) above for more details. **Q11:** Typo: $l$ not
appearing in equation (1). **A11:** Thanks for catching this typo. It should be $P$ instead of $l$.

[Meta-Review · NeurIPS 2020]

Three reviewers (two of which are knowledgeable) are in favor of acceptance. R2 is not convinced of the strength of the contribution, and suggests that a more thorough "Broader Impact" analysis of the approach in this paper --- on the other hand, s/he does not strongly object to this paper being accepted.